# Genome-wide CRISPR interference screen identifies long non-coding RNA loci required for differentiation and pluripotency

Jeffrey R. Haswell[1,2]☉, Kaia Mattioli[2,3]☉, Chiara Gerhardinger[3,4], Philipp G. Maass[5,6], Daniel J. Foster🄳[1,2], Paola Peinado[7], Xiaofeng Wang[8], Pedro P. Medina[7], John L. Rinn[9], Frank J. Slack🄳[1]*

**1** Department of Pathology, HMS Initiative for RNA Medicine, Beth Israel Deaconess Medical Center, Boston, Massachusetts, United States of America, **2** Department of Biological and Biomedical Sciences, Harvard Medical School, Boston, Massachusetts, United States of America, **3** Department of Stem Cell and Regenerative Biology, Harvard University, Cambridge, Massachusetts, United States of America, **4** Broad Institute of MIT and Harvard, Cambridge, Massachusetts, United States of America, **5** Genetics and Genome Biology Program, SickKids Research Institute, Toronto, Ontario, Canada, **6** Department of Molecular Genetics, University of Toronto, Toronto, Ontario, Canada, **7** Department of Biochemistry and Molecular Biology, University of Granada, Centre for Genomics and Oncological Research (GENYO), Granada, Spain, **8** Department of Molecular and Systems Biology, Geisel School of Medicine, Dartmouth College, Hanover, New Hampshire, United States of America, **9** Department of Biochemistry, University of Colorado, BioFrontiers Institute, Boulder, Colorado, United States of America

☉ These authors contributed equally to this work.
* fslack@bidmc.harvard.edu

**Data Availability Statement:** The accession number for the data reported in this paper is NCBI GEO: GSE137584. All code to reproduce analysis is

## Abstract

Although many long non-coding RNAs (lncRNAs) exhibit lineage-specific expression, the vast majority remain functionally uncharacterized in the context of development. Here, we report the first described human embryonic stem cell (hESC) lines to repress (CRISPRi) or activate (CRISPRa) transcription during differentiation into all three germ layers, facilitating the modulation of lncRNA expression during early development. We performed an unbiased, genome-wide CRISPRi screen targeting thousands of lncRNA loci expressed during endoderm differentiation. While dozens of lncRNA loci were required for proper differentiation, most differentially expressed lncRNAs were not, supporting the necessity for functional screening instead of relying solely on gene expression analyses. In parallel, we developed a clustering approach to infer mechanisms of action of lncRNA hits based on a variety of genomic features. We subsequently identified and validated *FOXD3-AS1* as a functional lncRNA essential for pluripotency and differentiation. Taken together, the cell lines and methodology described herein can be adapted to discover and characterize novel regulators of differentiation into any lineage.

## Introduction

The advent of deep sequencing technology has revealed the extensive and widespread nature of transcription across the human genome, with ~80% being transcribed at some point during

available at link: https://github.com/kmattioli/
2019__lncRNA_CRISPRi. All processed data are
available within the manuscript's Supporting
information files.

**Funding:** This work was funded by the US National
Institutes of Health (NIH) grant P01GM099117,
and by the National Science Foundation Graduate
Research Fellowship under the grant DGE1144152
(J.R.H. and K.M.). The funders had no role in study
design, data collection and analysis, decision to
publish, or preparation of the manuscript.

**Competing interests:** The authors have declared
that no competing interests exist.

development [1]. Given that less than 2% of the human genome encodes for protein-coding genes, there exists a vast and expanding network of non-coding RNAs, including long non-coding RNAs (lncRNAs), which are broadly defined as transcripts greater than 200 nucleotides in length with no apparent protein-coding function. Unlike other non-coding RNA families such as microRNAs and Piwi-interacting RNAs, lncRNAs display a wide variety of functions, acting as scaffolds, decoys, and guides in both the nucleus and cytoplasm [2].

Recently, a handful of lncRNAs have emerged as major regulators of key biological processes, including proliferation [3], pluripotency [4], and differentiation [5,6]. Expression profiling of lncRNAs during different stages of development has revealed highly tissue-specific and context-dependent expression, supporting their roles as potential developmental regulators [7]. Despite these preliminary observations, only a small fraction of lncRNAs have been characterized, both in terms of biological function as well as mechanism of action. Given that there are over 17,000 predicted lncRNA loci in the human genome [8], large-scale screens are necessary for elucidating functional lncRNAs. However, a robust, unbiased approach for genome-wide identification of developmentally relevant lncRNAs in human embryonic stem cells (hESCs) has yet to be described.

While the emergence of CRISPR gene editing has revolutionized scalable and cost-effective genetic screens, hESCs and lncRNAs both present a unique set of challenges for perturbing gene expression on a high-throughput scale. Notably, CRISPR editing in hESCs has proven more difficult than in immortalized cell lines, for several reasons. First, Cas9-induced double-stranded breaks have been shown to cause acute toxicity in hESCs in a P53-dependent manner [9]. In addition, hESCs have low transfection efficiencies [10] and demonstrate pervasive transgene silencing during development [11], making it difficult to stably express Cas9 throughout the course of differentiation.

Given their non-coding nature, lncRNAs are difficult to functionally disrupt using small deletions generated by traditional CRISPR-Cas9 methods [12]. Furthermore, while RNAi screens have identified biologically relevant lncRNAs during development [13], they exhibit more off-target effects [14], and are unable to effectively target lncRNAs localized to the nucleus [15]. CRISPR interference (CRISPRi) and CRISPR activation (CRISPRa) address the challenges of RNAi and represent effective tools for probing the function of lncRNAs [16]. These methods use catalytically-dead Cas9 (dCas9) proteins fused to effector domains to either repress (CRISPRi) or activate (CRISPRa) expression by targeting the transcription start site (TSS) of genes. While previous groups have constructed dCas9 stem cell lines that maintain expression during either endoderm [17], mesoderm [18], or ectoderm [19] differentiation, a dCas9-expressing hESC line that maintains stable targeting of both coding and non-coding genes during differentiation into all three primary germ layers has not yet been reported.

Here, we successfully developed CRISPRi and CRISPRa hESC lines that demonstrate robust repression and activation, respectively, of lncRNAs as well as protein-coding genes. These are the first reported CRISPR-dCas9 hESC lines to effectively modulate gene expression during differentiation into all three primary germ layers. We used our CRISPRi line to perform a genome-wide screen during definitive endoderm differentiation, identifying and validating several lncRNA loci that control pluripotency and regulate essential endoderm pathways. Furthermore, using our hits from the screen, we developed an unsupervised learning approach to predict the potential mechanisms of action of lncRNA loci based on select genomic features. Taken together, our study establishes an approach that can be scaled across multiple lineages, allowing for a high-throughput method for identifying functional, developmentally relevant lncRNAs.

## Materials and methods

### H1 hESC cell culture

We acquired H1 (WA01) hESCs from the University of Wisconsin (WiCell). We cultured H1 cells in mTeSR1 media (STEMCELL Technologies) on Matrigel-coated plates and changed media daily. We passaged the cells every 5–7 days, using Gentle Cell Dissociation Reagent (GCDR) or ReLeSR (STEMCELL Technologies).

### Generation of dCas9-KRAB/VP64 stable hESCs

To generate dCas9-KRAB-GFP and dCas9-VP64-GFP cell lines, we infected H1 hESCs with lentivirus containing dCas9-KRAB or dCas9-VP64 fused to GFP (via a T2A peptide cleavage linker). We obtained the dCas9-VP64-GFP construct from Addgene (#61422) and generated the dCas9-KRAB-GFP construct by cloning the KRAB cassette into the BamHI and NheI sites of the dCas9-VP64-GFP plasmid. Following infection, we sorted cells for GFP and plated sparsely (20,000 cells/well in 6-well plates) with 10 μM ROCK inhibitor (Y-27632) to obtain single-cell-derived isogenic colonies. Clones were tracked daily, picked manually using a P200 pipette tip, expanded, and tested for dCas9 expression via Western blot using a Cas9 antibody (CST; mouse mAb #14697). To generate the dox-inducible dCas9-KRAB-mCherry cell line, we nucleofected H1 hESCs with the pAAVS1-NDi-CRISPRi (Gen1) vector [20] along with *AAVS1* TALEN pair plasmids [20]. Following nucleofection, cells were selected for using 25 μg/mL Geneticin (Life Technologies), then seeded out and expanded using the same methods described above. dCas9-KRAB expression was induced by adding 2 μg/mL doxycycline to the mTeSR1 media.

### Endoderm, mesoderm, and ectoderm lineage induction

We used the STEMdiff Definitive Endoderm kit, STEMdiff Early Mesoderm kit, and SMADi Neural Induction kit to induce endoderm, mesoderm, and ectoderm differentiation, respectively. All kits are from STEMCELL Technologies, and differentiation was induced following the manufacturer's protocol.

### RT-qPCR

We isolated RNA using the Direct-zol Miniprep RNA kit (Zymo) with the DNase treatment step. RNA was quantified and reverse transcribed using oligo(dT)$_{20}$ primers and the Super-Script IV kit (ThermoFisher). RT-qPCR was performed on the LightCycler 480 II instrument (Roche). Reactions were performed in triplicate, and gene expression was normalized to *RPL19*. Error bars represent standard deviation of the mean. P values were calculated using an unpaired t-test.

### Flow cytometry and staining

hESCs and differentiated cells were harvested for staining or flow cytometry using GCDR. Cells were washed twice with 500 μL cold FBS stain buffer (BD #554656), fixed in 250 μL Cyto-fix fixation buffer (BD #554655) for 15 min at 4˚C, and washed twice with 500 μL cold FBS stain buffer. Cells were then stained (if necessary) with extracellular antibody markers for 30 min in the dark at 4˚C, washed twice with 500 μL cold FBS stain buffer. For subsequent intracellular staining, cells were washed twice with 500 μL cold 1X permeabilization buffer (BioLegend #421002), resuspended in 100 μL cold 1X permeabilization buffer, and incubated with intracellular antibody markers for 30 min in the dark at 4˚C. Prior to flow cytometry analysis, stained cells were washed twice with 500 μL cold 1X permeabilization buffer and resuspended

in 500 μL cold FBS stain buffer. A list of antibodies used in this study can be found in the STAR Methods. Cells were analyzed using the FACSymphony (BD), and sorted using the FACSAria (BD).

## lncRNA categorization

We downloaded the lncRNA annotation file from GENCODE v25 [8]. We then removed any lncRNAs found to have a conserved open reading frame longer than 100 amino acids via phyloCSF [21], thus assuring that we were analyzing *bona fide* non-coding RNAs. Finally, we classified lncRNAs into the following categories: (1) intergenic, defined as lncRNAs whose gene starts and ends were both > 1000 bp away from an annotated protein-coding gene, (2) promoter-overlapping, defined as lncRNAs whose gene starts were within 1000 bp of a protein-coding gene start, (3) transcript-overlapping, defined as lncRNAs whose transcripts overlapped a protein-coding transcript (given they were not promoter-overlapping), and (4) gene nearby, defined as lncRNAs that did not physically overlap any protein-coding genes, but had a protein-coding gene start or gene end within 1000 bp of its gene start or gene end.

## RNA-seq analysis

We isolated RNA from each lineage (two replicates each) using the Direct-zol Miniprep RNA kit (Zymo) with DNase treatment. We synthesized cDNA and performed Illumina paired end 75bp sequencing. We used the cutadapt program to trim adaptors and low-quality bases off of RNA-seq reads [22]. We then used kallisto [23] to pseudo-align and quantify reads to the GENCODE v25 transcriptome (in hg19 assembly coordinates). As *DIGIT* was not included in GENCODE v25, we manually added the transcript sequence that was described in Daneshvar et al. [5]. After reads were mapped, we performed differential expression analysis of transcripts using sleuth [24]. We separately tested for differential expression between hESCs and definitive endoderm and hESCs and early mesoderm using likelihood ratio tests. For gene-level quantifications, we summed all transcript counts for a given gene.

## Gene ontology enrichment analysis

Gene ontology (GO) enrichment analysis was obtained from the Gene Ontology Consortium [25,26], using the PANTHER Classification System (pantherdb.org). Significantly enriched pathways were determined using Fisher's Exact test with the Bonferroni correction for multiple testing.

## Tissue-specificity calculation

We calculated tissue-specificity across the three lineages (hESCs, endoderm, and mesoderm) using the *tau* metric [27], which has been shown to be robust to biases arising from low expression. *Tau* is calculated as follows:

$$\tau = \frac{\sum_{i=1}^{n}(1 - \hat{x}_i)}{n - 1}; \hat{x}_i = \frac{x_i}{\max_{1 \leq i \leq n}(x_i)}$$

where $x_i$ is the expression of a transcript in lineage $i$ and $n$ is the number of total lineages (here, 3). Thus, *tau* calculates the average difference between the expression of a transcript in a given lineage and its maximal expression across all lineages, meaning "ubiquitous" transcripts will have *tau* values close to zero while "lineage-specific" transcripts will have *tau* values close to one.

## TSS assignment

To assign TSSs to transcripts, we relied on the FANTOM5 CAGE-associated transcriptome (FANTOM-CAT) annotations [28], which sought to accurately define the 5' ends of human lncRNAs. If the transcript was explicitly assigned a TSS in FANTOM-CAT, as was the case for 46% of transcripts, we used the FANTOM-CAT assigned TSS. Otherwise, if the transcript was within 400 bp of an annotated CAGE TSS on the same strand, which was the case for an additional 26% of transcripts, we assigned the transcript the closest same-stranded CAGE TSS. For all other transcripts (28%), we used the GENCODE-annotated 5' end of the transcript as the TSS.

## CRISPRi library design

We included all 12,611 lncRNA transcripts expressed at $\geq$ 0.1 tpm in either hESCs, endoderm, or mesoderm. We also included a set of 24 literature-curated positive control transcripts that had been previously shown to be required for endoderm differentiation and 261 other highly differentially-expressed protein-coding genes (S2 Table). We used the GPP sgRNA Designer tool to design sgRNAs targeting the TSSs of the included transcripts (after re-assigning TSSs by the rules above) [29,30]. We assigned the top 10 sgRNAs to each TSS according to the ranking rules used by the GPP tool, which, briefly, prioritizes sgRNAs by distance to the TSS, a lack of off-targets, and predicted site chromatin accessibility [30]. To create negative control sgRNAs, we randomly sampled 500 of our designed sgRNAs and scrambled each one 10 times, making sure to eliminate any that we were able to BLAT to the human genome (parameters used: -stepSize = 5 -minScore = 0 -minIdentity = 0) for a total of 5000 scrambled sgRNAs. Finally, as some transcripts had the same TSS, or very close TSSs, we consolidated any duplicate sgRNAs in the library. All told, our CRISPRi library contained 111,801 unique sgRNAs. Of the 106,801 targeting sgRNAs, 93.2% were not predicted to have any off-targets near protein-coding TSSs, and 74.7% were not predicted to have any off-targets near non-coding gene TSSs, which is similar to previously described lncRNA-targeting CRISPRi libraries [31].

## CRISPRi library cloning

The sgRNA library was synthesized by Twist Biosciences. We then amplified the library using emulsion PCR and cloned it into the pCRISPRia-v2 vector as previously described [32]. We then packaged the library into lentivirus in HEK293T cells, also as previously described [32].

## CRISPRi endoderm differentiation screen

To perform the screen, we infected dCas9-KRAB-GFP cells with the lentivirally-packaged sgRNA library at an MOI < 0.3 and selected in 1 μg/mL puromycin. We then used the STEMdiff Definitive Endoderm kit (STEMCELL Technologies) to induce differentiation, according to the manufacturer's instructions. We isolated a sample of "Day Zero" cells to assess initial representation of sgRNAs. Five days post-differentiation induction, we harvested and stained cells for FOXA2 and SOX17, two markers of definitive endoderm. Specifically, we stained batches of 5 million cells with 20 μL FOXA2-APC (Miltenyi Biotec #130-107-774) and 4 μL SOX17-PE (Miltenyi Biotec #130-111-032). We used FACS to sort populations into undifferentiated (FOXA2/SOX17 double negative) and differentiated (FOXA2/SOX17 double positive) populations. We additionally sorted each population by BFP expression, as BFP was indicative of effective CRISPRi repression. For all populations, we isolated DNA from all cells using QIAamp columns (Qiagen), following the manufacturer's instructions and adding a de-crosslinking step (resuspended cells in 180 μL Buffer ATL and 20 μL proteinase K, incubated at

56˚C for 1 h, then 90˚C for 1 h, then proceeded with standard QIAamp protocol). Following DNA isolation, we PCR amplified the sgRNA region as previously described [32]. We prepared sequencing libraries as previously described [31], and performed high-throughput Illumina single end 50bp DNA sequencing to assess sgRNA counts. We performed two separate biological replicates of the entire screen.

### CRISPRi screen sgRNA filtering

We performed 3 consecutive filtering steps to remove noisy sgRNAs. First, we filtered sgRNAs to those that had ≥ 5 cpm in both Day Zero replicates (**S5C Fig in** S1 Appendix), resulting in a set of 76,091 sgRNAs. Then, we further filtered that set of sgRNAs to those that had ≥ 1 cpm in both undifferentiated replicates (Fig 3C), resulting in a set of 57,384 sgRNAs. Finally, we further filtered that set of sgRNAs, removing any sgRNAs targeting TSSs that did not have ≥ 3 sgRNAs that met the first two filters, resulting in a final set of 55,804 sgRNAs (Fig 3D).

### CRISPRi screen analysis

We calculated log2 foldchanges between undifferentiated counts and differentiated counts for all 111,801 sgRNAs using DESeq2 [33], including replicate as a term in the model to correct for batch effects. We then used CRISPhieRmix [34] on the DESeq2-calculated log2 foldchanges to find significant hits in our CRISPRi screen, after filtering the data as described in the previous section. We ran CRISPhieRmix with default parameters, using the filtered set of 2,690 scrambled sgRNAs as negative control inputs. As CRISPhieRmix only outputs an FDR per targeted TSS and no effect size, we estimated the effect size of each TSS to be the average of its top 3 most enriched (highest log2 foldchange) sgRNAs, as has been done previously [31].

### ENCODE expression and splicing efficiency quantification

We used ENCODE RNA-seq data to determine maximum transcript expression and splicing efficiency across a panel of 12 ENCODE cell lines (A549, GM12878, H1, HT1080, HUES64, IMR-90, K562, MCF-7, NCI-H460, SK-MEL-5, SK-N-DZ, and SK-N-SH). For both analyses, we downloaded the raw fastqs (after filtering out any samples that did not pass ENCODE's internal audits) and aggregated reads corresponding to technical replicates. For the expression analysis, we mapped the reads to the GENCODE v25 transcriptome (in hg19 assembly coordinates) using kallisto [23]. For the splicing efficiency analysis, we used kallisto to map reads to the same transcriptome, this time including all unspliced gene sequences as additional "transcripts". We calculated splicing efficiency for each gene as previously described; briefly, genes whose reads map entirely to the unspliced "transcript" will have a splicing efficiency of 0, whereas genes whose reads map only to spliced transcripts will have a splicing efficiency of 1 [35].

### Genomic feature aggregation

We aggregated a variety of genomic features for both lncRNAs and mRNAs. We aggregated the following features using GENCODE v25 annotations: (1) maximum RNA transcript length (i.e., length of the spliced transcript) across all transcripts for a gene, (2) maximum number of exons across all transcripts for a gene, (3) mean transcript GC content across all transcripts for a gene, and (4) DNA locus length (i.e., genomic distance from the transcript TSS to the 3' end of the transcript, which includes introns). We aggregated the following features using FANTOM5 data: (1) number of FANTOM5-annotated TSSs (active in any tissue) [36] within 100 bp of the GENCODE-annotated transcript TSS (on the same strand), (2) number of

FANTOM5-annotated enhancers (active in any tissue) [37] within 1 Mb of the transcript, and (3) distance from the TSS to the closest FANTOM5-annotated enhancer (active in any tissue). We aggregated the following features using the PhastCons 46-way sequence alignment [38]: (1) mean conservation score of the 200bp region surrounding the transcript TSS (maximum across all transcripts for a gene), and (2) mean conservation score across all exonic sequences in the transcript (maximum across all transcripts for a gene). We also determined the distance from the TSS to the closest endoderm-specific enhancer as defined by Loh et al., 2014 (their S5 Table) as well as the distance from the TSS to the closest definitive endoderm H3K27ac, H3K27me3, H3K4me2, and H3K4me3 peaks (peak calls downloaded from the Cistrome database) [39].

## GWAS

We downloaded the list of GWAS Catalog (all associations, v.1.0.2) from https://www.ebi.ac.uk/gwas/docs/file-downloads [40]. We then found the hg19 coordinates for each SNP using the 1000Genomes annotation [41]. We limited our analyses to SNPs associated with cancers of tissues that arise from the endoderm lineage (a list of traits used can be found in S3 Table). We then found the closest endoderm-cancer associated SNP to every gene.

## Feature-based clustering analysis

We normalized our data such that each feature had a mean of 1 and a standard deviation of 1 and performed k-means clustering on this highly dimensional standardized data (using 2 clusters and standard parameters in sklearn's KMeans function). We visualized results using a t-SNE plot (with 2 components and standard parameters in sklearn's TSNE function).

# Results

## Generation of CRISPRi and CRISPRa hESC lines that stably express dCas9 throughout differentiation into all three germ layers

We sought to develop the tools necessary to perform genome-wide perturbation screens throughout differentiation. To this end, we established CRISPR interference (CRISPRi) and CRISPR activation (CRISPRa) systems in human embryonic stem cells (hESCs) that could effectively repress or activate, respectively, gene targets in all three primary germ layers: ectoderm, endoderm, and mesoderm. Specifically, we infected H1 (WA01) hESCs with lentiviral GFP constructs containing EF1-α promoter-driven dCas9 fused to either a Krüppel-associated box (KRAB) repressor domain (Fig 1A) or a VP16 tetramer (VP64) activation domain (**S1A Fig in** S1 Appendix). We chose the EF1-α promoter due to its diminished propensity for silencing in embryonic stem cells [42], and screened hundreds of isogenic clones for constitutive expression of dCas9-KRAB or dCas9-VP64 during early differentiation. We identified clones that stably expressed dCas9 in an undifferentiated state after several passages, as well as throughout the course of definitive endoderm and early mesoderm differentiation (Fig 1B and **S1B Fig in** S1 Appendix), as well as neural progenitor cell (ectoderm) differentiation (Fig 1C and **S1C Fig in** S1 Appendix). Fluorescence-activated cell sorting (FACS) analysis of our clones showed >99% GFP expression in each germ layer (Fig 1D and **S1D Fig in** S1 Appendix), suggesting uniform and stable dCas9 expression. In addition to our constitutively-expressed clones, we used a TALEN-assisted gene-trap approach to insert a dox-inducible dCas9-KRAB-mCherry cassette into the *AAVS1* safe harbor locus in H1 cells (**S2A Fig in** S1 Appendix). Similarly, this isogenic hESC line displayed expression of dCas9-KRAB in each primary germ layer (**S2B-S2D Fig in** S1 Appendix).

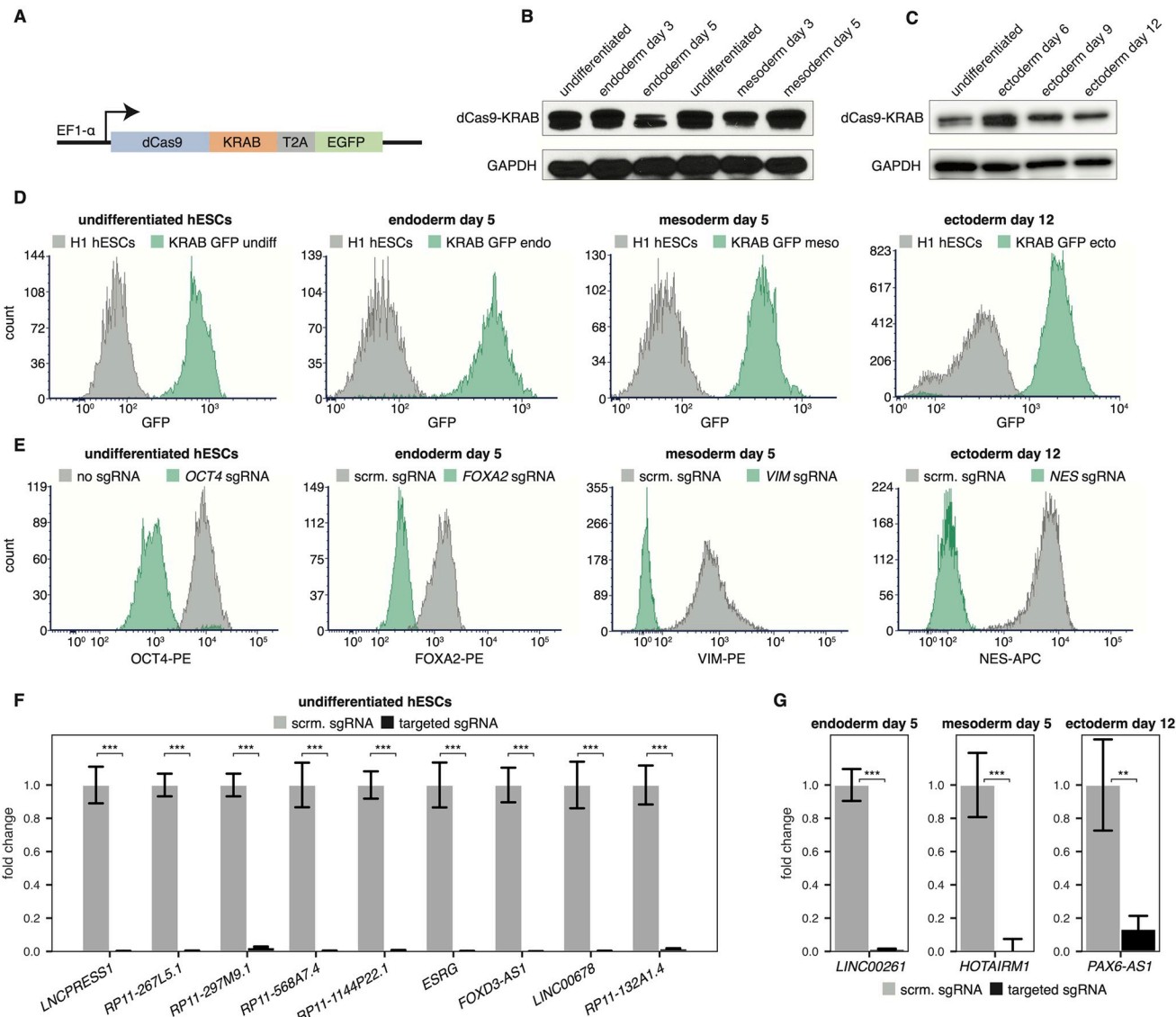

**Fig 1. Establishing a CRISPRi cell line that maintains repression during differentiation. (A)** Lentiviral GFP construct containing dCas9-KRAB driven by the EF1-α promoter. **(B)** Western blot of dCas9-KRAB over the course of definitive endoderm and early mesoderm differentiation. **(C)** Western blot of dCas9-KRAB over the course of neural progenitor cell (ectoderm) differentiation. **(D)** FACS analysis of GFP expression for undifferentiated and differentiated dCas9-KRAB-GFP line compared to H1 cells. **(E)** FACS staining of targeted mRNA genes in undifferentiated and differentiated dCas9-KRAB-GFP cells. **(F)** RT-qPCR expression of targeted lncRNAs in undifferentiated dCas9-KRAB-GFP hESCs. *** = p < 0.001 by an unpaired t-test. **(G)** RT-qPCR expression of targeted lncRNAs during endoderm, mesoderm, and ectoderm differentiation in dCas9-KRAB-GFP cells. ** = p < 0.01, *** = p < 0.001 by an unpaired t-test.

After confirming stable expression of dCas9-KRAB and dCas9-VP64 in our isogenic lines, we evaluated the knockdown and activation efficiency of our CRISPRi and CRISPRa systems, respectively. We cloned single guide RNAs (sgRNAs) into the pCRISPRia-v2 expression vector [43], targeting the transcription start site (TSS) of both protein-coding genes and long non-coding RNA (lncRNAs). After lentiviral sgRNA transduction, puromycin selection, and induction of differentiation of our isogenic lines, we observed robust knockdown (Fig 1E–1G and **S2E Fig in** S1 Appendix) or activation (**Fig S1E in** S1 Appendix) of targeted mRNAs and lncRNAs, as measured by FACS staining and RT-qPCR, respectively. We consistently

measured knockdown levels greater than 99%, and activation levels that exceeded 4-fold upregulation.

Taken together, these isogenic lines represent the first reported hESC cell lines to stably express dCas9-KRAB or dCas9-VP64 and effectively repress or activate target genes throughout the course of differentiation into the three primary germ layers. Thus, we have successfully engineered cell lines which can be employed to perform genome-wide gene perturbation throughout differentiation.

## Genome-wide profiling of lncRNAs during early differentiation into endoderm and mesoderm

We next sought to define a subset of candidate lncRNAs that may be acting in definitive endoderm differentiation based on gene expression patterns. To this end, we performed RNA-sequencing on two biological replicates from undifferentiated hESCs, day 5 definitive endoderm, and day 5 early mesoderm (which is derived from the same mesendodermal lineage as definitive endoderm), and quantified lncRNA and mRNA expression at the transcript and gene levels across all samples (Fig 2A and **S3A Fig in** S1 Appendix; see Materials and methods).

We confirmed that expected markers were highly expressed in their respective lineages, such as *POU5F1* (*OCT4*) and *NANOG* in hESCs, *EOMES* and *GATA6* in definitive endoderm, and *T* in early mesoderm (Fig 2B). Gene ontology analysis of differentially expressed genes in endoderm or mesoderm compared to undifferentiated hESCs revealed enrichment of the expected biological pathways (Fig 2C; see methods).

We found that 8,190 lncRNA genes corresponding to 12,611 lncRNA transcripts were expressed at $\geq 0.1$ tpm in at least one of the three lineages (undifferentiated hESCs, endoderm, or mesoderm) (Fig 2D and **S3B Fig in** S1 Appendix; S1 Table). While 3,702 (29%) of these lncRNA transcripts were intergenic (further than 1000bp away from a protein-coding transcript), the majority of these lncRNA transcripts were either very close to or overlapping protein-coding genes (Fig 2D). Additionally, 4,158 lncRNA transcripts were differentially expressed between either hESCs and definitive endoderm or hESCs and mesoderm (Fig 2E). The majority of these lncRNA transcripts (2,939 (71%)) were uniquely differentially expressed in only one of the lineages. On average, lncRNAs were more specifically expressed than mRNAs when examining expression profiles across the three lineages (Fig 2F; see methods). Moreover, lncRNAs were also more specifically expressed than known human transcription factors (as defined by [44]) across these three lineages (Fig 2F).

Collectively, these data represent a robust overview of both the coding and non-coding transcriptome in hESCs, definitive endoderm, and early mesoderm, and define a subset of lncRNAs that are putatively functioning in hESC differentiation.

## CRISPRi screen to identify lncRNA loci essential for proper differentiation

We next used our RNA-seq data to design a comprehensive CRISPRi screen of lncRNAs expressed during endoderm differentiation. Specifically, we included all 12,611 lncRNA transcripts expressed at $\geq 0.1$ tpm in either undifferentiated hESCs, definitive endoderm, or early mesoderm. We additionally included a set of 24 literature-curated positive control transcripts (e.g. *FOXA2*, *SOX17*, *GATA6*, *EOMES*) that are known to be required for endoderm differentiation, as well as 261 additional highly differentially expressed protein-coding transcripts. The full list of transcripts targeted by our library can be found in S2 Table.

CRISPRi induces knockdown by targeting a transcriptional repressor domain (KRAB) to the TSSs of genes. Because TSSs are often misannotated for lowly-expressed transcripts [28],

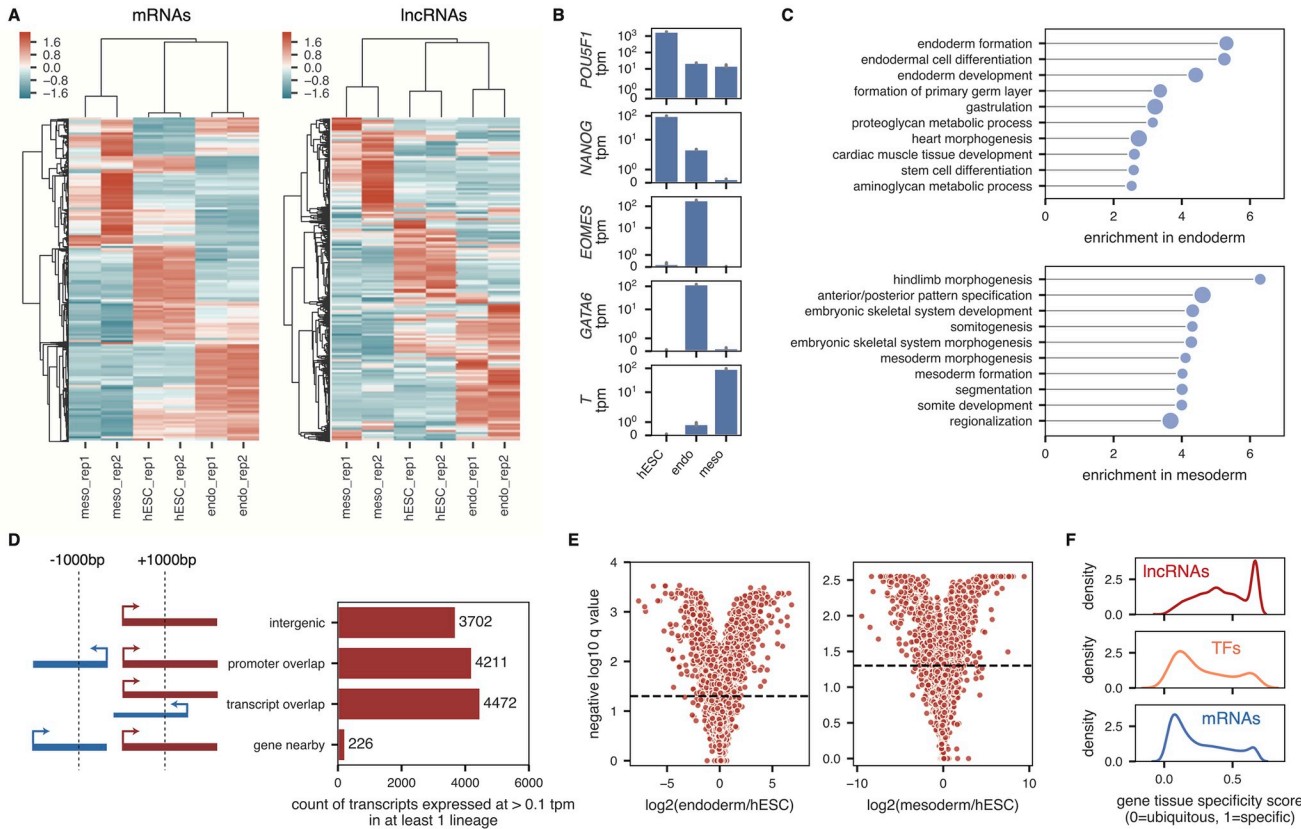

**Fig 2. lncRNAs are differentially and specifically expressed across hESCs, definitive endoderm, and early mesoderm. (A)** Heatmap showing the expression, standardized per sample (column z-score) for the 10,000 most highly expressed protein-coding genes (left) and all 8,190 lncRNA genes with a minimum gene tpm of 0.1 in at least 1 sample (right). Genes (rows) and samples (columns) are hierarchically clustered using the correlation as the distance metric. **(B)** Average expression in each lineage of lineage marker genes *POU5F1* (*OCT4*), *NANOG*, *EOMES*, *GATA6*, and *T*. Gray bars correspond to 90% confidence intervals, calculated across 2 biological replicates. **(C)** Top 10 gene ontology terms significantly enriched in either endoderm (top) or mesoderm (bottom) compared to undifferentiated hESCs. Size of dots is inversely proportional to the Bonferroni-corrected p-value; all plotted terms have p-values < 0.05. **(D)** Count of transcripts expressed at a minimum of 0.1 tpm in either hESCs, endoderm, or mesoderm, broken up into either protein-coding genes (blue) or lncRNAs (red). The lncRNAs are further classified based on their genomic proximity to other transcripts, as outlined in the schematic to the left. **(E)** Volcano plots showing the log2 expression fold-change between endoderm and hESCs (left) and mesoderm and hESCs (right) for all lncRNA transcripts. Horizontal lines define a q-value cut-off of 0.05. **(F)** Tissue-specificity of genes calculated across hESCs, endoderm, and mesoderm, for all lncRNA genes (top) and all mRNA genes (bottom) as well as a set of curated human transcription factors (middle) from Lambert et al. [44]. Gene-level tpms were used.

we refined the annotated TSSs of genes in our screen using a set of high-confidence TSSs defined by the FANTOM5 consortium [45] (**Fig S3C and S3D in** S1 Appendix; see methods). We consolidated these TSSs (some of which corresponded to multiple transcripts) into a list of 10,852 unique TSSs. We then designed 10 single guide RNAs (sgRNAs) targeting each of these TSSs using the optimized GPP sgRNA Designer tool, which minimizes off-target effects while maximizing knockdown efficiency [29,30]. Finally, we sampled 500 of these targeting sgRNAs and randomly scrambled them 10 times to create a set of 5,000 non-targeting negative control sgRNAs. In total, after removing duplicates, our CRISPRi sgRNA library included 111,801 unique sgRNAs (Fig 3A).

We transduced the lentiviral pooled CRISPRi library into undifferentiated dCas9-KRAB-GFP hESCs and selected with puromycin. After 4–5 days of selection and expansion (**S4A Fig in** S1 Appendix), we isolated cells from this "Day Zero" population to assess the overall sgRNA distribution (Fig 3B). We then induced definitive endoderm differentiation using

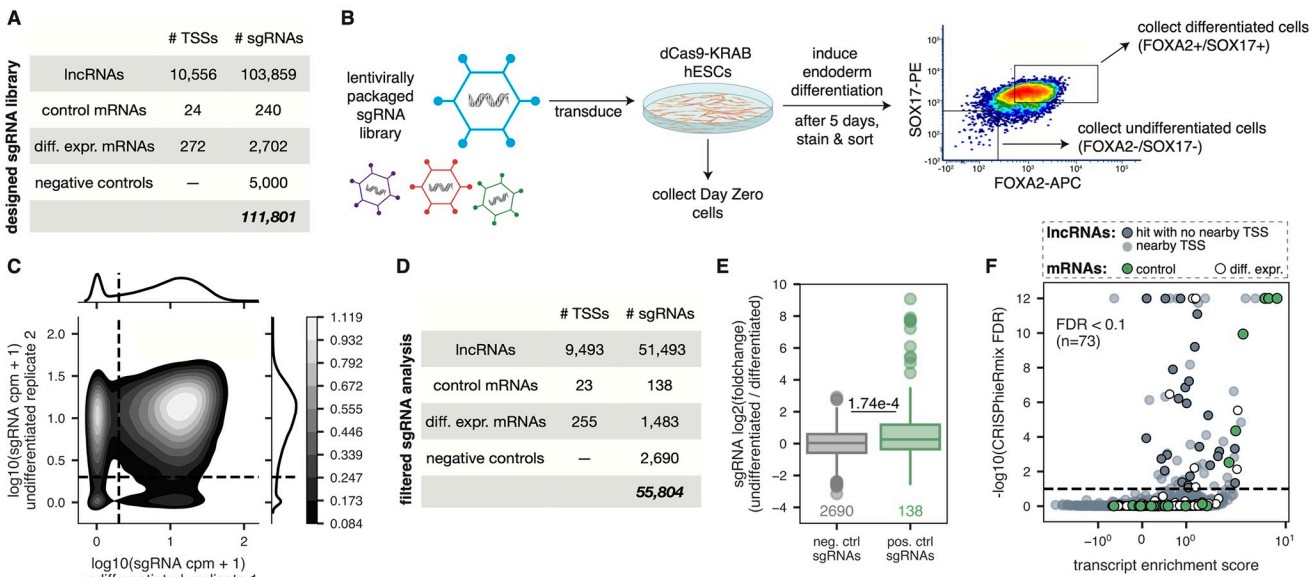

**Fig 3. CRISPRi screen identifies lncRNA loci that reproducibly affect endoderm differentiation. (A)** Number of TSSs targeted and corresponding unique sgRNAs in the CRISPRi library. **(B)** Schematic overview of the CRISPRi screen. The sgRNA library was packaged into lentivirus and transduced into dCas9-KRAB-GFP hESCs, which were then induced to undergo endoderm differentiation. Day Zero cells were collected to measure initial representation of the sgRNA library. After five days, the cells were sorted for endoderm markers SOX17 and FOXA2, and populations of differentiated and undifferentiated cells were collected, as shown in the representative FACS plot. PE = phycoerythrin. APC = allophycocyanin. **(C)** Kernel density estimate plot showing the counts per million in the undifferentiated populations for both replicate 1 and replicate 2. Horizontal and vertical lines define a cpm of 1 in each replicate; color bar indicates densities of sgRNAs. **(D)** Number of TSSs targeted and corresponding unique sgRNAs after filtering steps that are used in downstream analyses. **(E)** Distribution of sgRNA enrichments (DESeq2 log2 fold-change of undifferentiated counts compared to differentiated counts, including a batch term) for scrambled negative control sgRNAs (gray) and positive control targeting sgRNAs (green). P-value shown is from a one-sided Mann Whitney test. **(F)** Plot showing the transcript enrichment score (average DESeq2 log2 fold-change of top 3 most enriched sgRNAs) and CRISPhieRmix global FDR values for each lncRNA transcript (gray), positive control transcript (green), and differentially expressed mRNA transcript (white) tested. Hit lncRNA transcripts that do not have another gene TSS within 1000 bp of their own TSS are further highlighted (dark gray). Horizontal line defines an FDR cut-off of 0.1.

the STEMdiff Definitive Endoderm Kit (see methods). After five days, we stained cells for FOXA2 and SOX17 expression—two markers of definitive endoderm—and used FACS to sort high-confidence populations of FOXA2+/SOX17+ and FOXA2-/SOX17- cells (Fig 3B and **S4B-S4D Fig in** S1 Appendix). Finally, we characterized the sgRNA representation in each of these populations using targeted sequencing.

To ensure reproducibility of our results and limit false positives, we performed two independent biological replicates of the screen. In each replicate, endoderm differentiation was extremely efficient and undifferentiated cells were sorted conservatively (**S4D and S4E Fig in** S1 Appendix). As expected, replicates were generally correlated, with most noise presenting as sampling noise among low count sgRNAs in the undifferentiated replicates (**S5A Fig in** S1 Appendix).

We next performed two important filtering steps to address the inherent noise of the screen. First, we required sgRNAs to have $\geq 5$ counts per million (cpm) in both Day Zero replicates to limit any noise due to poor initial sgRNA representation (**S5B Fig in** S1 Appendix). Second, since we gated FOXA2-/SOX17- undifferentiated cells conservatively, a subset of sgRNAs were present in only one of the two undifferentiated population replicates (Fig 3C). Thus, we required sgRNAs to have $\geq 1$ cpm in both undifferentiated replicates. Finally, we required that targeted TSSs have $\geq 3$ sgRNAs that meet these criteria. Together, these filtering

steps resulted in a set of 55,804 reproducibly represented sgRNAs targeting 9,771 TSSs (90% of all TSSs included in the library) (Fig 3D and **S5C and S5D Fig in** S1 Appendix).

We then used DESeq2 to calculate the enrichment (i.e. log2 fold change of counts) of these 55,804 filtered sgRNAs in the undifferentiated FOXA2-/SOX17- populations compared to the differentiated FOXA2+/SOX17+ populations. Given the sgRNA count variance that was introduced by performing two conservative biological replicates of the screen (**S5E Fig in** S1 Appendix), we included a term for batch (replicate) in the DESeq2 model when calculating log2 fold changes. As expected, sgRNAs targeting positive controls had significantly higher log2 fold changes than scrambled sgRNAs (Fig 3E).

To further validate the quality of our screen, we compared our results to a previous lncRNA-targeting growth screen performed in iPSC cells (CRiNCL) [31]. We reasoned that sgRNAs targeting lncRNAs that affect iPSC growth, as found by CRiNCL, should affect growth in hESCs as well, and therefore drop out of our differentiated samples. Indeed, among the 1117 sgRNAs that overlapped between our screen and CRiNCL, we found that sgRNA drop out correlated between the two screens (**S5F Fig in** S1 Appendix).

To determine which TSSs were highly likely to contribute to proper endoderm differentiation, we used CRISPhieRmix, a tool which was specifically developed to address issues unique to CRISPRi/a screens, such as variable sgRNA efficiency [34]. The Bayesian hierarchical mixture modelling strategy employed by CRISPhieRmix allows for more statistical power in large-scale pooled CRISPRi screens while controlling false discovery rates [46]. We confirmed that our data met the assumptions employed by CRISPhieRmix—namely, that the majority of targeting sgRNAs resemble negative control sgRNAs (**S5G Fig in** S1 Appendix) and that sgRNAs targeting positive control genes follow a mixture distribution composed of only a subset of sgRNAs that work (**S5H Fig in** S1 Appendix). We then applied CRISPhieRmix to the set of 55,804 filtered sgRNAs (S3 Table; see Materials and methods).

In total, 73 TSSs out of the 9,771 TSSs tested were significantly enriched in the undifferentiated population at an FDR < 0.1 (Fig 3F). As expected, many of the most significant hits were positive control mRNAs, including *SOX17*, *FOXA2*, *EOMES*, and *GATA6*. However, it is important to note that several positive control genes were not recovered in the screen. Thus, while our conservative FACS gating resulted in a list of 73 TSSs that are reproducibly and significantly associated with proper endoderm differentiation, we lack the power to infer the number of putatively functional lncRNAs due to a high false negative rate. Of the 73 significant TSSs, 60 corresponded to lncRNA loci, and 26 of these lncRNA TSSs were not within 1000 bp of another gene TSS (Fig 3F).

Collectively, these data show that our genome-wide CRISPRi screen was able to reproducibly recapitulate expected biological signal, and suggest that, at minimum, dozens of lncRNA loci may be required for proper endoderm differentiation. We provide the results of our screen for all targeted lncRNAs and mRNAs as a resource to the community (S2 and S3 Tables).

## Validation of individual screen hits

Following our CRISPRi screen, we validated several control (mRNA) hits, lncRNA loci hits, and lncRNA loci non-hits (S4 Table). To do this, we selected 22 sgRNAs present in the screening library and tested them each individually. All sgRNAs were cloned and subsequently transduced into our dCas9-KRAB-GFP hESC line using the pCRISPRia-v2 expression vector [32]. Following puromycin selection, we induced infected cells into definitive endoderm for five days, then stained for FOXA2/SOX17 to assess differentiation efficiency. In addition, we performed RT-qPCR to determine knockdown efficiency of each individual sgRNA.

For each individually tested sgRNA, we calculated a validation enrichment score based on the ratio of undifferentiated cells to differentiated cells five days post endoderm differentiation. We observed a significant (p = 0.002) correlation between screen sgRNA enrichment score and the validation enrichment score (Fig 4A).

We observed significant knockdown for the majority of our targets (Fig 4B and 4C; S4 Table), confirming the efficiency and robustness of our dCas9-KRAB-GFP cell line. As expected, sgRNAs targeting mRNA hits (e.g. *SOX17*, *EOMES*) resulted in severe loss-of-differentiation phenotypes (Fig 4E; S4 Table), as demonstrated by reduction of SOX17 and/or FOXA2 expression at the protein level (via FACS). While knockdown of our lncRNA loci hits resulted in more subtle phenotypes, we observed consistent reduction of endoderm markers compared to scrambled sgRNA controls (Fig 4F; S4 Table). Interestingly, targeted knockdown of certain lncRNA loci resulted in reduced expression of one specific marker (e.g. reduced SOX17 expression in *RP11-222K16.2* knockdown), suggesting a potential mechanism of regulation. Taken together, we were able to individually validate mRNA and lncRNA loci hits resulting from our genome-wide CRISPRi screen. While we validated a subset of hits and non-hits, further work is needed to validate the full set of lncRNA loci targets described in this study.

## Common features of lncRNA hits

After validating the results of the screen, we next sought to determine what features, if any, were common to the hit lncRNA loci that were required for differentiation (Fig 5A and **S6 Fig in** S1 Appendix; see methods). Given the potentially high false negative rate of our screen, we focused on comparing our hit lncRNA loci to a set of "stringent non-hit" lncRNA loci: targeted lncRNA TSSs with $\geq 9$ filtered sgRNAs represented and a CRISPhieRmix FDR of $> 0.9$. All told, our list of stringent non-hit lncRNA loci contained 158 unique genes (S5 Table). We found that our hit lncRNA loci were significantly closer to known endoderm cancer-associated SNPs than stringent non-hit lncRNA loci (Fig 5B; S5 and S6 Tables). In particular, three of our hit lncRNA loci—*RP11-867G2.8*, *RP11-541P9.3*, and *VLDLR-AS1*—contain endoderm cancer-associated SNPs in their gene bodies. While *VLDLR-AS1* is known to play important roles in liver and esophageal cancer [47,48], *RP11-541P9.3* and *RP11-867G2.8* remain uncharacterized.

We found that our hit lncRNA loci had significantly more nearby FANTOM5-defined TSSs (on the same strand) than stringent non-hit lncRNA loci (Fig 5C). Our hit lncRNAs had significantly higher expression in our endoderm RNA-seq data than stringent non-hit lncRNAs (Fig 5D). Hit lncRNAs also had higher maximum expression values across a panel of 12 ENCODE cell lines (Fig 5F) and were more efficiently spliced than stringent non-hit lncRNAs across this same panel of ENCODE cell lines (Fig 5E; see methods).

Interestingly, while expression levels and differential expression levels were higher in hit lncRNAs compared to stringent non-hit lncRNAs, we did not observe a significant enrichment of hits among differentially expressed transcripts (two-sided Fisher's exact test odds ratio = 1.27, p-value = 0.36). Indeed, the proportion of hit lncRNAs to stringent non-hit lncRNAs is similar within differentially expressed and non-differentially expressed transcripts (19% in both cases) (Fig 5G). Thus, lncRNA expression profiles are not necessarily predictive of biological function of lncRNA loci. Taken together, these data suggest that while some features are associated with hits, loss-of-function screens remain pivotal in assessing the functionality of lncRNA loci.

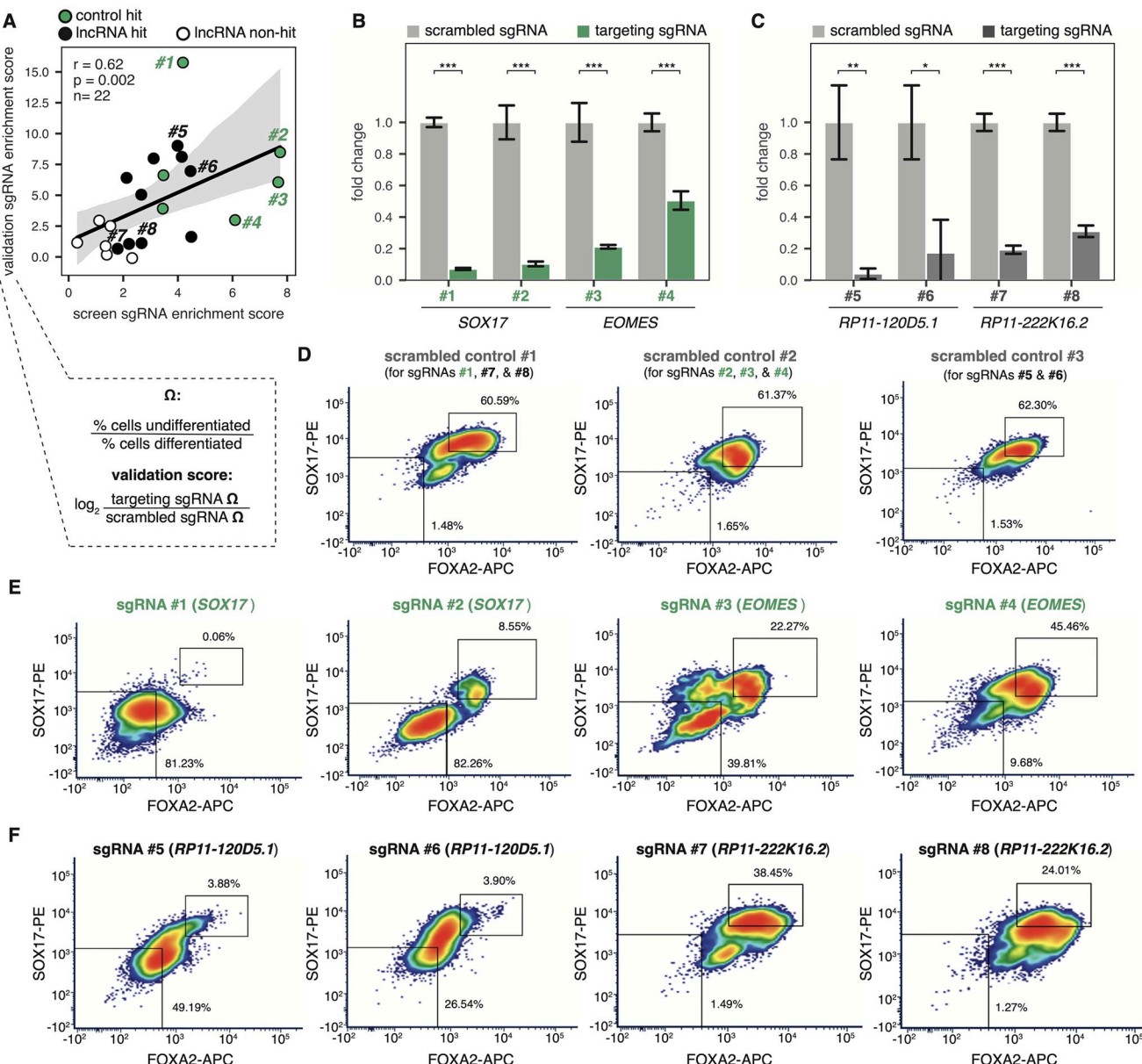

**Fig 4. CRISPRi screen reveals several lncRNA loci required for endoderm differentiation. (A)** Correlation of the validation sgRNA score and the screen sgRNA score for all 22 sgRNAs validated, colored by whether the sgRNAs target control mRNA hits (green), lncRNA hits (black), or lncRNA non-hits (white). Validation scores were calculated according to the formula outlined. Example validation data for 8 sgRNAs targeting 2 control mRNA and 2 lncRNA hits are shown in (B)-(F); the full dataset is in S4 Table. r = Spearman's rho, p = Spearman p-value, n = # sgRNAs. **(B)** RT-qPCR expression of mRNA *SOX17* (sgRNAs #1 and #2) or *EOMES* (sgRNAs #3 and #4) during endoderm differentiation using either a scrambled negative control sgRNA (light gray bars) or a targeting sgRNA (green bars). Labeled sgRNA numbers correspond to those outlined in (A). *** = p-value < 0.001 by an unpaired t-test. **(C)** RT-qPCR expression of lncRNA *RP11-120D5.1* (sgRNAs #5 and #6) or *RP11-222K16.2* (sgRNAs #7 and #8) during endoderm differentiation using either a scrambled negative control sgRNA (light gray bars) or a targeting sgRNA (dark gray bars). Labeled sgRNA numbers correspond to those outlined in (A). * = p-value < 0.05, ** = p-value < 0.01, *** = p-value < 0.001 by an unpaired t-test. **(D)** FACS staining of scrambled negative control sgRNAs during endoderm differentiation, with percentages of differentiated cells (top right box) and undifferentiated cells (bottom left box) labeled. Three separate endoderm differentiation time courses were performed to validate the 8 outlined sgRNAs. Scrambled sgRNA #1 was run as a negative control for targeting sgRNAs #1, #7, and #8; scrambled sgRNA #2 was run as a negative control for targeting sgRNAs #2, #3, and #4; scrambled sgRNA #3 was run as a negative control for targeting sgRNAs #5 and #6. **(E)** FACS staining of mRNA-targeting sgRNAs #1–4 during endoderm differentiation, with percentages of differentiated cells (top right box) and undifferentiated cells (bottom left box) labeled. **(F)** FACS staining of lncRNA-targeting sgRNAs #5–8 during endoderm differentiation, with percentages of differentiated cells (top right box) and undifferentiated cells (bottom left box) labeled.

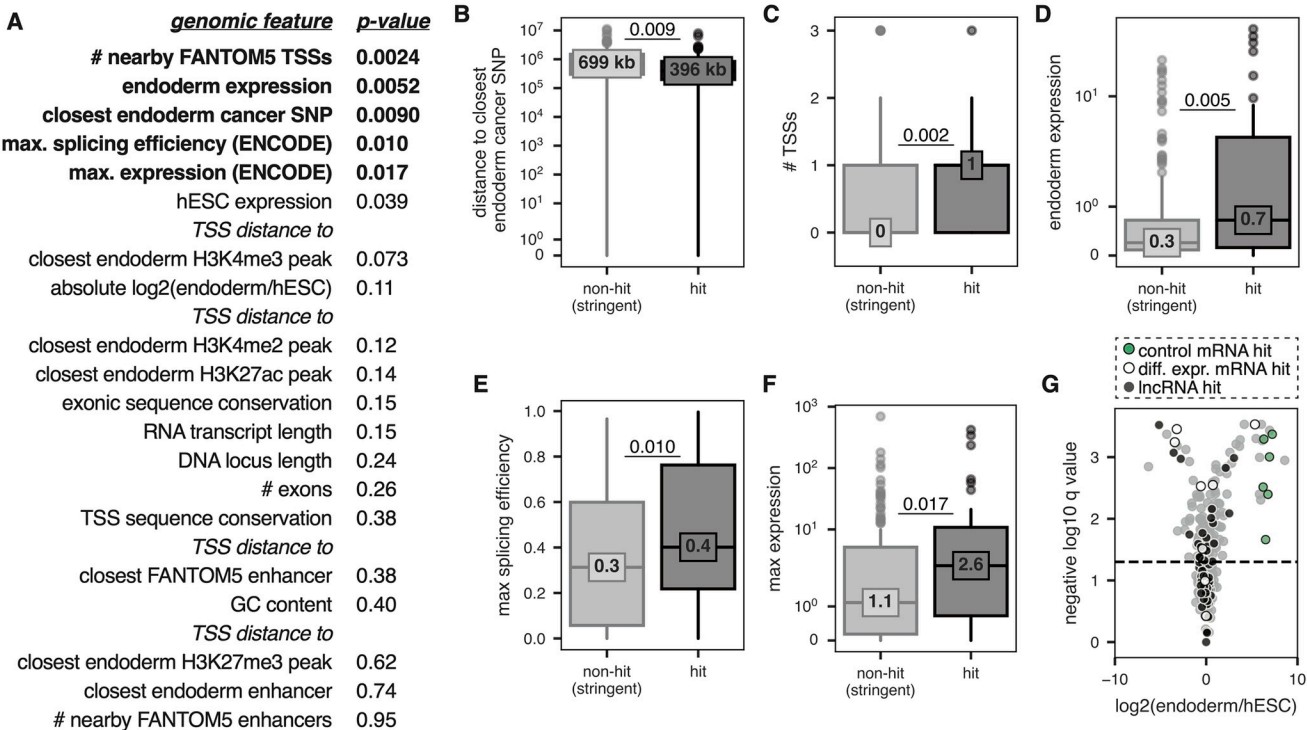

**Fig 5. lncRNA hits are associated with certain genomic features. (A)** List of genomic features analyzed and their associated two-sided Mann Whitney *p*-values comparing hit lncRNAs vs. stringent non-hit lncRNAs. Features with Benjamini-Hochberg adjusted FDRs of < 0.10 are shown in bold. **(B-F)** Boxplots showing the distributions of various genomic features outlined in (A) for hit lncRNAs and stringent non-hit lncRNAs. Median value of each distribution is also shown. *P*-values are from two-sided Mann Whitney tests. **(G)** Volcano plot showing the RNA-seq results for stringent non-hits (in gray, all biotypes) and hits (colored by biotype as shown above the plot). Horizontal line depicts an FDR cut-off of 0.05 for significant differential expression between endoderm and hESCs.

## Inferring the mechanisms of action of hit lncRNA loci

As CRISPRi induces the formation of heterochromatin around a guide RNA's target site, any DNA regulatory activity that the target locus may possess is shut down along with RNA transcription [49]. Thus, using CRISPRi, it is impossible to distinguish lncRNAs acting through

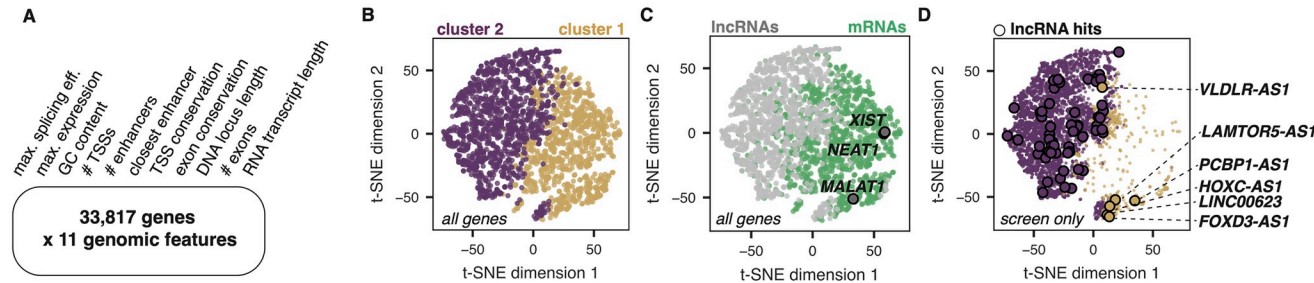

**Fig 6. Using k-means clustering of 11 genomic features to find candidate functional RNA molecules. (A)** List of the 11 genomic features used in clustering analysis across 33,817 genes. **(B)** t-SNE visualization of the data matrix outlined in (A), colored by k-means cluster assignment. Plot shows a sub-sample of the data: 1000 randomly-sampled lncRNAs and 1000 randomly-sampled mRNAs. **(C)** Same t-SNE as in (A), now colored by biotype, where gray dots are lncRNAs and green dots are mRNAs. The 3 "gold standard" lncRNAs (*XIST*, *NEAT1*, and *MALAT1*) are highlighted and outlined in black. **(D)** Subset of the same t-SNE in (A) and (B), now showing only lncRNAs included in the screen, colored by their cluster assignment. Significant hits in either cluster are outlined in black. Hits assigned to cluster 1 are annotated.

DNA-based mechanisms (e.g. as enhancers) from lncRNAs acting through RNA-based mechanisms (e.g. as scaffolds or decoys). We sought to predict which of our lncRNA loci hits, if any, were most likely to have RNA-based mechanisms of action.

While the number of characterized lncRNAs is growing, there remain very few whose mechanisms of action are well-characterized. We therefore could not rely on supervised learning approaches to perform mechanistic predictions, as we currently lack a trusted set of training data. We previously showed that known functional lncRNAs with likely RNA-based mechanisms resemble mRNAs across a variety of genomic features including expression patterns, splicing efficiency, and conservation [35]. Thus, we used unsupervised methods to cluster genes based on a set of 11 general genomic features (Fig 6A; see Materials and methods).

We used k-means to cluster this highly dimensional dataset into 2 clusters and visualized the results using the t-distributed stochastic neighbor embedding (t-SNE) dimensionality reduction approach (Fig 6B). As expected, cluster 1 contained primarily lncRNAs while cluster 2 contained primarily mRNAs (Fig 6C). While the data exists on a continuum of all considered features, cluster separation was driven primarily by splicing efficiency, maximum expression, and exon conservation (**S7 Fig in** S1 Appendix). Three "gold standard" lncRNAs known to exhibit RNA-based mechanisms (*XIST*, *NEAT1*, and *MALAT1*) all clustered with mRNAs (Fig 6C), which is consistent with our previous work [35]. We found that while 6 lncRNA hits clustered with mRNAs (*VLDLR-AS1*, *PCBP-AS1*, *LAMTOR5-AS1*, *HOXC-AS1*, *LINC00623*, and *FOXD3-AS1*), the majority of our lncRNA hits did not (Fig 6D). Moreover, for each of these 6 lncRNAs, there is experimental evidence for an RNA-based phenotype, either through lncRNA knockdown approaches (*VLDLR-AS1* and *LAMTOR5-AS1*) [47,48,50] or, in 4 cases (*HOXC-AS1*, *FOXD3-AS1*, *PCBP-AS1* and *LINC00623*), also including lncRNA ectopic overexpression approaches [51–54].

Taken together, these data suggest that most lncRNA hits resulting from our CRISPR screen are likely acting as DNA regulatory elements, but a subset may be playing a role in endoderm differentiation through RNA-based mechanisms. To facilitate future validation work, the results of our feature-based clustering analysis are available in S7 Table.

## Characterization of *FOXD3-AS1* as a regulator of pluripotency and endoderm pathways

Finally, we investigated the potential function of one our lncRNA hits that we predicted to have an RNA-based mechanism of action, *FOXD3-AS1* (CRISPhieRmix FDR = 0.026). Of the 6 lncRNA gene hits predicted to have RNA-based mechanisms of action, *FOXD3-AS1* showed the highest level of differential expression between endoderm and hESCs in our RNA-seq data (Fig 7A). However, *FOXD3-AS1* follows an unexpected expression pattern: it is one of the most highly enriched lncRNAs in undifferentiated hESCs and demonstrates significant down-regulation during endoderm and mesoderm differentiation (Fig 7B; S1 Table). Given this unexpected expression pattern, we were interested in elucidating the role of *FOXD3-AS1* in potentially regulating pluripotency and differentiation.

Because the *FOXD3-AS1* promoter overlaps the promoter of *FOXD3* (a transcription factor required for self-renewal of stem cells), we used multiple shRNAs to specifically target the lncRNA (Fig 7C). While maintaining hESCs in stem cell media, visible differentiation occurred (Fig 7D), with RT-qPCR and FACS analysis confirming complete loss of pluripotency markers at 18 days post infection (**S8A Fig in** S1 Appendix). We also observed strong upregulation of several key endoderm factors (Fig 7E), including *GATA6*, *FOXA2*, *EOMES*, and *CXCR4* in response to *FOXD3-AS1* knockdown. Following upregulation of endoderm factors, we observed an eventual loss of pluripotency markers (**S8A Fig in** S1 Appendix). A time

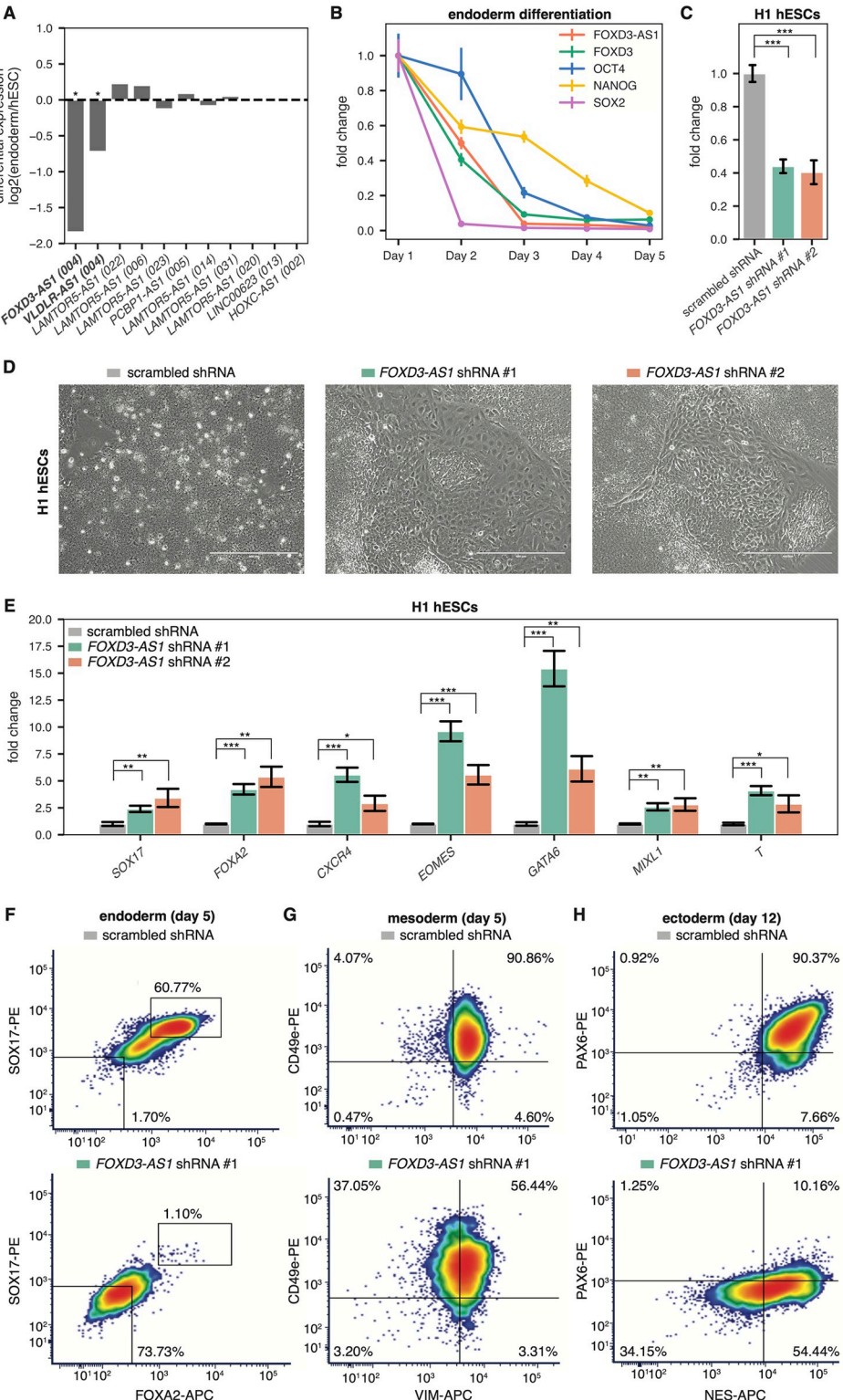

**Fig 7. *FOXD3-AS1* expression and knockdown in hESCs.** **(A)** Log2 fold change between expression in hESCs and endoderm for all hit lncRNA transcripts in the screen whose loci were predicted to have RNA-based mechanisms of action (Fig 6). * = significant differential expression, sleuth q-value < 0.05. The *LAMTOR5-AS1* hit TSS targeted in our screen could not resolve between 6 different *LAMTOR5-AS1* transcripts (GENCODE transcript numbers in parantheses), and as such they are all plotted here. **(B)** RT-qPCR expression time course during definitive endoderm

differentiation of H1 hESCs. **(C)** RT-qPCR expression of *FOXD3-AS1* 21 days post infection of H1 hESCs with *FOXD3-AS1* shRNAs. *** = p < 0.001 by an unpaired t-test. **(D)** Phase-contrast images of H1 hESCs infected with *FOXD3-AS1* shRNAs. Images were taken 21 days post infection. **(E)** RT-qPCR expression of pluripotency/differentiation genes 21 days post infection of H1 hESCs with *FOXD3-AS1* shRNAs. * = p < 0.05, ** = p < 0.01, *** = p < 0.001 by an unpaired t-test. **(F-H)** FACS staining of day 5 definitive endoderm cells (F), day 5 early mesoderm cells (G), or day 12 neural progenitor cells (H) infected with scrambled shRNA or *FOXD3-AS1* shRNA #1. Cells were fixed and stained with antibodies against FOXA2 and SOX17 (F), VIM and CD49e (G), or NES and PAX6 (H).

course knockdown of *FOXD3-AS1* (**S8B Fig in** S1 Appendix) showed early induction of *EOMES*, along with reduction of *FOXD3* expression within 48 h. To test whether *FOXD3-AS1* was required for differentiation into other lineages, we induced early mesoderm or ectoderm differentiation in hESCs following knockdown of *FOXD3-AS1*. We observed significant reduction in properly differentiated cells in each lineage using two different shRNAs (Fig 7F–7H and **S8C-S8E Fig in** S1 Appendix), suggesting that loss of *FOXD3-AS1* results in dysregulation of pluripotency pathways required for early embryogenesis.

Following knockdown experiments, we cloned and overexpressed *FOXD3-AS1* in undifferentiated hESCs and observed reduced expression of several endoderm factors (**S8F Fig in** S1 Appendix), consistent with the role of *FOXD3-AS1* as repressing endoderm pathways. Taken together, these experiments support *FOXD3-AS1* as a novel pluripotency factor required for pluripotency via repression of the endoderm lineage. Moreover, our RNA knockdown results support the prediction that *FOXD3-AS1* exerts its function through RNA-based mechanisms. Our results are in agreement with a recent publication [55] which found that the *FOXD3-AS1* lncRNA is localized to the cytoplasm and affects pluripotency of hESCs through modulating Wnt signaling, an essential pathway required for definitive endoderm differentiation [56]. Thus, the CRISPRi screening approach outlined herein, coupled with additional bioinformatic analyses, revealed developmentally relevant lncRNA loci, a subset of which likely act as functional RNAs.

## Discussion

Here, we report the first described set of hESC cell lines that can maintain dCas9 expression (and successfully modulate target gene expression) throughout differentiation into all three germ layers (Fig 1, and **S1** and **S2 Figs in** S1 Appendix). These cell lines can be used for CRISPRi screens, as we have shown in this work, as well as CRISPRa screens. Although we focus on early endoderm differentiation in this study, we also show that these cell lines maintain dCas9 expression throughout mesoderm and ectoderm differentiation. Thus, the methodology described can be adapted to discover novel regulators (both coding and non-coding) of differentiation into any lineage. The cells we have engineered are available to the community upon request. Additionally, we provide all data produced in this work—including the RNA-seq data, the results of the CRISPRi screen, and the clustering analysis—as resources to the community.

Our CRISPRi screen reproducibly identified known endoderm factors as top hits, including *FOXA2* and *SOX17*, as well as *EOMES*, *GATA6*, and *GSC*. Moreover, we identified 73 lncRNA loci, including the previously reported endoderm regulator *DIGIT* [5]. We used a conservative approach to identify hits: we performed two independent biological replicates of the screen and performed conservative FACS gating of differentiated and undifferentiated populations. We speculate that we have missed a number of functional lncRNAs, due to both the conservative nature of our screen, as well as the fact that lncRNAs are known to often exert quite subtle effects on gene expression [57]. Nevertheless, our approach yielded a list of dozens of lncRNA loci that are likely required for proper endoderm differentiation (S3 Table), most of which are entirely uncharacterized. Further validation and functional studies are needed to characterize

the specific mechanisms by which these lncRNA loci regulate endoderm differentiation, whether via direct regulation of endoderm factors (e.g. *DIGIT*) or via regulation of pluripotency (e.g. *FOXD3-AS1*).

Of these lncRNA hits, only 16 of them are intergenic (greater than 1000 bp away from any other gene). Thus, the majority of lncRNA hits we identified are either physically overlapping a protein-coding gene or within 1000 bp of a protein-coding gene promoter; this proximity makes it likely that some sgRNAs in our screen were also affecting expression of the protein-coding genes proximal to these lncRNAs. However, there is precedence for this class of lncRNAs to play *bona fide* roles in early development. Indeed, divergent lncRNAs (which are included in our "promoter overlap" biotype) have been found to play particularly important roles in pluripotency and differentiation [58]. In addition, a literature analysis of lncRNA hits near protein-coding gene TSSs revealed that only 10 out of 34 overlapped with known regulators of hESC differentiation (**S9 Fig in** S1 Appendix), providing a high confidence list of potential protein-coding genes primed for further investigation.

Using CRISPRi to knockdown lncRNA loci precludes us from inferring RNA-based mechanisms of action for hits, as the heterochromatin recruited by dCas9-KRAB will shut down DNA regulatory elements as well as transcription. In order to infer lncRNAs with likely RNA-based mechanisms of action, we aggregated a variety of genomic datasets and performed unsupervised clustering. Based on previous work [35], we expected lncRNAs with RNA-based mechanisms to cluster with mRNAs. Using this method, we predicted that only 6 of our screen hits were likely to exhibit RNA-based mechanisms of action (Fig 6D). We note that it is difficult to robustly evaluate our approach without more gold standard examples of lncRNAs that act as *bona fide* functional RNAs, and we also note that lncRNAs can have multiple mechanisms of action. As additional data becomes available, more sophisticated models will likely be necessary, and experimental validation of predictions will always be essential. However, our clustering approach provides a foundational set of candidate functional lncRNAs to prioritize for future validation (S7 Table).

We validated a top hit from our screen, *FOXD3-AS1*, and found that it is required for maintaining hESC pluripotency. *FOXD3-AS1* is significantly differentially expressed between hESCs and endoderm, but surprisingly, it is upregulated ~300-fold in hESCs compared to endoderm. Our data show that *FOXD3-AS1* is required for pluripotency by acting as a repressor of endoderm factors. We speculate that repression of *FOXD3-AS1* at the incorrect developmental time point results in aberrant expression of endoderm factors, which disrupts the pluripotent state of hESCs. Subsequent downregulation of stemness factors (*OCT4*, *NANOG*, *SOX2*) leads to loss of pluripotency (S8A Fig in S1 Appendix), which ultimately results in improper differentiation into endoderm, as well as other lineages. Despite having promoter overlap with the known pluripotency regulator *FOXD3*, shRNAs targeting the *FOXD3-AS1* transcript mimicked the sgRNA phenotypes observed in the screen, pointing towards an RNA-based mechanism of action, as our k-means clustering predicted (Fig 6D).

Finally, our work underscores the importance of performing functional screens to characterize lncRNAs. While we found that some features were associated with lncRNA hits (Fig 5), most features examined were not significantly different between hits and non-hits. Importantly, we also observed that expression levels are not necessarily predictive of biological function, as the proportion of hit lncRNA loci was roughly equal between loci harboring differentially expressed lncRNAs and non-differentially expressed lncRNAs (Fig 5H). Thus, loss-of-function screens are paramount to characterizing the vast non-coding transcriptome and its role in development and differentiation. Taken together, our CRISPRi and CRISPRa cell lines described herein can serve as a resource for functional screens in any of the three

primary germ layers, identifying and elucidating the role of both coding and non-coding genes in development.

## Supporting information

**S1 Table. RNA-seq lncRNA expression in hESCs, endoderm, and mesoderm.**
(TXT)

**S2 Table. CRISPRi screen sgRNA counts in each population.**
(TXT)

**S3 Table. CRISPhieRmix results per targeted transcript in screen.**
(TXT)

**S4 Table. Validation results for the 22 sgRNAs tested individually.**
(TXT)

**S5 Table. Genomic features of all lncRNAs targeted in the screen.**
(TXT)

**S6 Table. List of cancers/traits from endoderm-derived tissues.**
(TXT)

**S7 Table. Clustering results of all annotated lncRNAs and mRNAs.**
(TXT)

**S1 Appendix. Document containing S1-S9 Figs.**
(DOCX)

**S1 Raw images. Labeled unadjusted/uncropped images for all blots/gels included in manuscript.**
(PDF)

## Acknowledgments

The authors thank members of the Slack lab for technical help (Allison Baker, Christos Miliotis, Emanuelle Grody, Jihoon Lim, Soomi Lee, Tanvi Saxena); the Harvard Medical School Department of Immunology's Flow Cytometry Core Facility for technical support (Chad Araneo, Felisha Lopez, Daniela Gutierrez); and Scott Younger, Martin Sauvageau, Mo Mandegar, Max Horlbeck, Kaveh Daneshvar, and Ryan Genga for advice and guidance.

## Author Contributions

**Conceptualization:** Jeffrey R. Haswell, Kaia Mattioli, John L. Rinn, Frank J. Slack.

**Data curation:** Jeffrey R. Haswell, Kaia Mattioli.

**Formal analysis:** Jeffrey R. Haswell, Kaia Mattioli.

**Funding acquisition:** Pedro P. Medina, John L. Rinn.

**Investigation:** Jeffrey R. Haswell, Kaia Mattioli, Chiara Gerhardinger, Philipp G. Maass, Daniel J. Foster, Paola Peinado, Xiaofeng Wang.

**Methodology:** Jeffrey R. Haswell, Kaia Mattioli, Chiara Gerhardinger, Philipp G. Maass, Xiaofeng Wang, John L. Rinn, Frank J. Slack.

**Project administration:** Jeffrey R. Haswell, Kaia Mattioli, John L. Rinn, Frank J. Slack.

**Resources:** John L. Rinn, Frank J. Slack.

**Software:** Kaia Mattioli.

**Supervision:** Pedro P. Medina.

**Validation:** Jeffrey R. Haswell.

**Visualization:** Jeffrey R. Haswell, Kaia Mattioli.

**Writing – original draft:** Jeffrey R. Haswell, Kaia Mattioli, John L. Rinn, Frank J. Slack.

**Writing – review & editing:** Jeffrey R. Haswell, Kaia Mattioli, Chiara Gerhardinger, Philipp G. Maass, Pedro P. Medina, John L. Rinn, Frank J. Slack.

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
