## [Decision Letter · Decision Letter 0]

9 Jul 2021

PONE-D-21-17843

Genome-wide CRISPR interference screen identifies long non-coding RNA loci required for differentiation and pluripotency

PLOS ONE

Dear Dr. Slack,

Thank you for submitting your manuscript to PLOS ONE. After careful consideration, we feel that it has merit but does not fully meet PLOS ONE’s publication criteria as it currently stands. Therefore, we invite you to submit a revised version of the manuscript that addresses the points raised during the review process.

We look forward to receiving your revised manuscript.

Kind regards,

Romi Gupta

Academic Editor

PLOS ONE

Journal Requirements:

Reviewers' comments:

Reviewer's Responses to Questions

**Comments to the Author**

1. Is the manuscript technically sound, and do the data support the conclusions?

Reviewer #1: Yes

Reviewer #2: Yes

2. Has the statistical analysis been performed appropriately and rigorously? 

Reviewer #1: Yes

Reviewer #2: Yes

3. Have the authors made all data underlying the findings in their manuscript fully available?

Reviewer #1: Yes

Reviewer #2: Yes

4. Is the manuscript presented in an intelligible fashion and written in standard English?

Reviewer #1: Yes

Reviewer #2: Yes

5. Review Comments to the Author

Reviewer #1: The authors have performed well designed and well controlled experiments in the manuscript.

I do not have any suggestions for major experiments for the manuscript.

Two minor suggestions:

1. Hit lncRNA and non hit lncRNA do they have any specific chromatin signature, as authors suggest a major portion of hit lncRNA acts through DNA elements.

2. FOXD3 coding gene promoter do show any change at local chromatin structure in the shRNA KD?

Reviewer #2: In this study “Genome-wide CRISPR interference screen identifies long non-coding RNA loci required for differentiation and pluripotency” author developed CRISPRi and CRISPRa hESC lines and demonstrated robust repression and activation, respectively, of lncRNAs as well as protein-coding genes that effectively modulate gene expression that involved hESC lines. Author also developed an unsupervised learning approach to predict the potential mechanisms of action of lncRNA loci based on select genomic features. The current study successfully established an approach that can be applied to multiple lineage specific differentiation processes.

Minor comments.

1. Figure.1 B. the represented western blots are saturated, please replace with another exposure.

2. Figure 1. author encouraged to show the expression of differentiation markers as primary evidence of the model systems.

3. Figure 7C. Author encouraged to show the knockdown FOXD3-AS1 efficiency by western blotting.

4. Author encouraged to study one more lncRNA from Figure 7A.

6. PLOS authors have the option to publish the peer review history of their article (what does this mean?). If published, this will include your full peer review and any attached files.

Reviewer #1: No

Reviewer #2: **Yes: **Suresh Chava

---

## [Author Response · Author response to Decision Letter 0]

16 Aug 2021

Full response to reviewers is attached in the manuscript data as it contains a plot, but is copy-pasted here for convenience.

Reviewer #1: The authors have performed well designed and well controlled experiments in the manuscript.

I do not have any suggestions for major experiments for the manuscript.

Two minor suggestions:

1. Hit lncRNA and non hit lncRNA do they have any specific chromatin signature, as authors suggest a major portion of hit lncRNA acts through DNA elements.

We thank the reviewer for this suggestion. While we did look to see if our hit lncRNA loci were closer to nearby endoderm enhancers than non-hit loci (Figure S6K), we did not initially look at specific chromatin marks. We have therefore downloaded the ChIP-seq peaks for the 4 chromatin marks studied in definitive endoderm in Loh et al. 2014—H3K27ac, H3K27me3, H3K4me3, and H3K4me2—and determined their locations in relation to our hit lncRNA loci vs. non-hit lncRNA loci. We found that hit lncRNAs were, on average, closer than non-hit lncRNA loci to definitive endoderm H3K4me3 peaks, H3K4me2 peaks, and H3K27ac peaks, though all had only moderate Mann Whitney p-values (minimum 0.073, maximum 0.136). Thus, both promoter marks (H3K4me3 and H3K4me2) and enhancer marks (H3K27ac) are somewhat enriched near hit lncRNA loci, consistent with our conclusion that many of the lncRNA loci may act as DNA elements. The only ChIP-seq profile which did not show any differences between hits and non-hits is H3K27me3, which is expected, as this mark delineates silenced (not actively expressed) regions.

We have added the following plots to Figure S6:

And have updated the Methods section to describe the above analysis: “We also determined the distance from the TSS to the closest endoderm-specific enhancer as defined by Loh et al., 2014 (their Table S5) as well as the distance from the TSS to the closest definitive endoderm H3K27ac, H3K27me3, H3K4me2, and H3K4me3 peaks (peak calls downloaded from the Cistrome database) (Liu et al., 2011).”

2. FOXD3 coding gene promoter do show any change at local chromatin structure in the shRNA KD?

While we did not examine the local chromatin structure of the FOXD3 coding gene promoter upon FOXD3-AS1 shRNA KD, our group is currently exploring the mechanism behind FOXD3-AS1 regulation of pluripotency, including examining changes in chromatin landscape upon FOXD3-AS1 KD. Given that similar phenotypes were observed in both DNA- (CRISPRi) and RNA-based (shRNA) methods of FOXD3-AS1 KD, we believe that FOXD3-AS1 acts as a functional lncRNA as opposed to a DNA element. This hypothesis is supported by the recent study by Guo et al. (2020; cited in our manuscript) that characterized FOXD3-AS1 as acting in the cytoplasm to block the interaction between �-TrCP and phosphorylated �-catenin, leading to suppressed �-catenin degradation and WNT activation. We are currently performing ChIP-seq experiments to examine potential chromatin changes mediated at the FOXD3/FOXD3-AS1 promoter locus and are investigating the role that FOXD3 may play in FOXD3-AS1-mediated control of pluripotency.

Reviewer #2: In this study “Genome-wide CRISPR interference screen identifies long non-coding RNA loci required for differentiation and pluripotency” author developed CRISPRi and CRISPRa hESC lines and demonstrated robust repression and activation, respectively, of lncRNAs as well as protein-coding genes that effectively modulate gene expression that involved hESC lines. Author also developed an unsupervised learning approach to predict the potential mechanisms of action of lncRNA loci based on select genomic features. The current study successfully established an approach that can be applied to multiple lineage specific differentiation processes.

Minor comments.

1. Figure.1 B. the represented western blots are saturated, please replace with another exposure.

We thank the reviewer for this suggestion. Unfortunately there are no additional Western blot exposures that were taken, so we are unable to provide those images without performing additional experiments. All authors on this manuscript are no longer in the corresponding authors’ labs, making it difficult to repeat the experiments in a timely manner. However, we are confident that the conclusions from Figure 1B are supported by several additional data points in the paper, including Figure 1D demonstrating maintenance of GFP expression throughout differentiation in all three lineages and Figures 1E-1G demonstrating robust knockdown of both protein-coding (1E) and lncRNA (1F-G) genes. If dCas9-KRAB expression was not maintained over the course of endoderm and mesoderm differentiation, the levels of target gene repression would likely not be as strong.

2. Figure 1. author encouraged to show the expression of differentiation markers as primary evidence of the model systems.

We thank the reviewer for this suggestion. The levels of pluripotency and differentiation markers observed in our engineered lines (both protein-coding and non-coding genes) were similar to those observed in wild-type H1 hESCs, as evidenced by high RT-qPCR expression of lncRNAs in the absence of targeted sgRNAs (Figure 1F-1G), as well as strong protein expression observed by FACS during the screens (Figure S4) and in individual validation experiments (Figure 4).

3. Figure 7C. Author encouraged to show the knockdown FOXD3-AS1 efficiency by western blotting.

We thank the reviewer for the suggestion. Since FOXD3-AS1 does not code for a protein, we are unable to demonstrate knockdown efficiency via Western blotting, and instead showed RT-qPCR to demonstrate efficient knockdown of the RNA transcript.

4. Author encouraged to study one more lncRNA from Figure 7A.

We thank the reviewer for the suggestion. The main purpose of this report is to serve as a resource for the community rather than a deep dive into specific lncRNAs. We are currently investigating several lncRNAs identified from the screen (in addition to FOXD3-AS1) for their role in mediating pluripotency and differentiation. We look forward to sharing those findings in a future publication.

---

## [Editor Report · Decision Letter 1]

27 Aug 2021

Genome-wide CRISPR interference screen identifies long non-coding RNA loci required for differentiation and pluripotency

PONE-D-21-17843R1

Dear Dr. Slack,

We’re pleased to inform you that your manuscript has been judged scientifically suitable for publication and will be formally accepted for publication once it meets all outstanding technical requirements.

Kind regards,

Romi Gupta

Academic Editor

PLOS ONE
---

## [Editor Report · Acceptance letter]

25 Oct 2021

PONE-D-21-17843R1 

Genome-wide CRISPR interference screen identifies long non-coding RNA loci required for differentiation and pluripotency 

Dear Dr. Slack:

I'm pleased to inform you that your manuscript has been deemed suitable for publication in PLOS ONE. Congratulations! Your manuscript is now with our production department. 

Kind regards, 

on behalf of

Dr. Romi Gupta 

Academic Editor

PLOS ONE